# Lack of Evidence for the Role of the p.(Ser96Ala) Polymorphism in Histidine-Rich Calcium Binding Protein as a Secondary Hit in Cardiomyopathies

**DOI:** 10.3390/ijms242115931

**Published:** 2023-11-03

**Authors:** Stephanie M. van der Voorn, Esmée van Drie, Virginnio Proost, Kristina Dimitrova, Robert F. Ernst, Cynthia A. James, Crystal Tichnell, Brittney Murray, Hugh Calkins, Ardan M. Saguner, Firat Duru, Patrick T. Ellinor, Connie R. Bezzina, Sean J. Jurgens, J. Peter van Tintelen, Toon A. B. van Veen

**Affiliations:** 1Department of Medical Physiology, Division Heart & Lungs, University Medical Center Utrecht, 3584 CM Utrecht, The Netherlandsk.dimitrova@umcutrecht.nl (K.D.); 2Department of Genetics, Division Heart & Lungs, University Medical Center Utrecht, 3584 CM Utrecht, The Netherlands; 3Netherlands Heart Institute, 3511 EP Utrecht, The Netherlands; 4Departments of Clinical and Experimental Cardiology, Amsterdam Cardiovascular Sciences, Amsterdam University Medical Centers Location Academic Medical Center, 1105 AZ Amsterdam, The Netherlandsc.r.bezzina@amsterdamumc.nl (C.R.B.);; 5Department of Medicine, Division Cardiology, Johns Hopkins University, Baltimore, MD 21218, USActichne1@jhmi.edu (C.T.);; 6Department of Cardiology, University Heart Center Zurich, University Hospital Zurich, 8091 Zurich, Switzerland; 7Center for Integrative Human Physiology (ZIHP), University of Zurich, 8091 Zurich, Switzerland; 8Cardiovascular Disease Initiative, Broad Institute of MIT and Harvard, Cambridge, MA 02142, USA

**Keywords:** arrhythmia, cardiomyopathies, genetic modifier, heart failure, genetics

## Abstract

Inherited forms of arrhythmogenic and dilated cardiomyopathy (ACM and DCM) are characterized by variable disease expression and age-related penetrance. Calcium (Ca^2+^) is crucially important for proper cardiac function, and dysregulation of Ca^2+^ homeostasis seems to underly cardiomyopathy etiology. A polymorphism, c.286T>G p.(Ser96Ala), in the gene encoding the histidine-rich Ca^2+^ binding (HRC) protein, relevant for sarcoplasmic reticulum Ca^2+^ cycling, has previously been associated with a marked increased risk of life-threatening arrhythmias among idiopathic DCM patients. Following this finding, we investigated whether p.(Ser96Ala) affects major cardiac disease manifestations in carriers of the phospholamban (*PLN*) c.40_42delAGA; p.(Arg14del) pathogenic variant (cohort 1); patients diagnosed with, or predisposed to, ACM (cohort 2); and DCM patients (cohort 3). We found that the allele frequency of the p.(Ser96Ala) polymorphism was similar across the general European–American population (control cohort, 40.3–42.2%) and the different cardiomyopathy cohorts (cohorts 1–3, 40.9–43.9%). Furthermore, the p.(Ser96Ala) polymorphism was not associated with life-threatening arrhythmias or heart failure-related events across various patient cohorts. We therefore conclude that there is a lack of evidence supporting the important role of the *HRC* p.(Ser96Ala) polymorphism as a modifier in cardiomyopathy, refuting previous findings. Further research is required to identify bona fide genomic predictors for the stratification of cardiomyopathy patients and their risk for life-threatening outcomes.

## 1. Introduction

Inherited cardiomyopathies represent relatively rare but morbid diseases and may lead to severe heart failure (HF), life-threatening ventricular arrhythmias (VA), and even sudden cardiac death (SCD) at a young age [1]. The prevalence of inherited cardiomyopathies is estimated at 1/250 to 1/500 for hypertrophic cardiomyopathy (HCM), 1/250 for dilated cardiomyopathy (DCM), and 1/5000 for arrhythmogenic cardiomyopathy (ACM) [2,3,4,5]. In the majority of cases, an autosomal dominant inheritance can be found, yet recessive (homozygous or compound heterozygous) or X-linked forms can be identified [6,7]. Rare pathogenic variants are identified in a large proportion of cardiomyopathy patients, including genes encoding cardiac desmosomal proteins for arrhythmogenic cardiomyopathy (ACM), genes encoding sarcomeric proteins for hypertrophic cardiomyopathy (HCM) and dilated cardiomyopathy (DCM), and genes encoding structural myocardial proteins in DCM [4,8,9]. Clinical and genetic overlap between those cardiomyopathies is recognized, which may compromise an accurate diagnosis. For example, the pathogenic variant c.40_42delAGA; p.(Arg14del) in the phospholamban (*PLN*) gene may manifest as ACM or DCM (OMIM 609909) [10].

Disease onset and severity are highly variable. Even within families carrying the same genetic variant, variable disease expression and age-related incomplete penetrance are frequently observed [1]. Therefore, additional genetic and non-genetic factors (secondary hits) are believed to contribute to disease development and progression. For example, intensive exercise might increase disease penetrance and enhance the risk for life-threatening VA in ACM patients [11]. Similarly, the presence of multiple (likely) pathogenic variants in ACM-associated genes may increase disease severity [12]. Nevertheless, the prediction of severe outcomes among ACM and DCM patients remains exceedingly difficult, necessitating the identification of novel and accurate markers for risk stratification.

Recent and significant focus has been on the disturbance of calcium (Ca^2+^) homeostasis within cardiomyocytes with regard to cardiomyopathy etiology. Ca^2+^ plays a crucial role in the excitability and contraction of the heart. In failing cardiomyocytes, disturbed Ca^2+^ handling contributes to HF and life-threatening VA [13]. Therefore, dysfunction of factors that regulate Ca^2+^ homeostasis, such as the histidine-rich calcium-binding protein (HRC), may contribute to the development and progression of inherited cardiomyopathies [14]. HRC can interact with triadin when [Ca^2+^] levels increase, thereby regulating RyR2 and modulating SR Ca^2+^ release. When [Ca^2+^] is low in the SR, HRC binds to the sarcoplasmic/endoplasmic reticulum Ca^2+^-ATPase (SERCA2a), which suppresses SERCA2a function [14,15]. The serine residue at position 96 is a phosphorylation site of potential importance to triadin affinity (Figure 1). In one candidate gene study, a polymorphism in the *HRC* gene, c.286T>G p.(Ser96Ala)), was found to confer a four-fold higher risk of life-threatening VA and SCD in a small cohort of 123 patients with idiopathic DCM [16].

To follow up on this observation, the aim of our study was to examine (1) the presence of the *HRC* p.(Ser96Ala) polymorphism in the general population (control cohort); (2) the effect of the *HRC* p.(Ser96Ala) polymorphism on major cardiac events in three patient cohorts, namely carriers of the *PLN* p.(Arg14del) pathogenic variant (cohort 1); patients diagnosed with, or predisposed to, ACM (cohort 2); and DCM patients (United Kingdom (UK) Biobank, cohort 3). We hypothesized, based on the earlier observation, that the *HRC* p.(Ser96Ala) polymorphism may contribute to the risk of major disease manifestation, which could potentially improve risk stratification.

## 2. Results

### 2.1. Patient Characteristics

In cohort 1, 1005 carriers of the *PLN* p.(Arg14del) variant passed quality control (QC) and were included. Due to incomplete or unavailable health record data, 157 patients were excluded, leaving 848 patients for data analysis. In cohort 2, among 1033 ACM patients of European (-American) ancestry, 882 patients with complete health record data were included for further analysis. Finally, 1031 DCM patients of European ancestry were found in cohort 3, of which 985 individuals passed QC and were used for downstream analyses (Figure 2).

Table 1, Table 2 and Table 3 summarize the clinical characteristics of the specific patient cohorts *PLN* p.(Arg14del), ACM, and DCM, respectively, dividing the cohorts into wild-type (TT), heterozygous (TG) or homozygous (GG) for the *HRC* polymorphism. The median age of cohort 1 (*PLN* p.(Arg14del) carriers) ranged between 49 (interquartile range (IQR) 32–62) years for homozygous, 51 (IQR 37–63) for the wild type (WT), and 52 (IQR 36–65) years for heterozygous *HRC* carriers. No significant differences in male sex (43% for the WT; 45% for heterozygous and homozygous) or index patients (WT = 20%, heterozygous = 19%, and homozygous = 16%) were found between the three different polymorphism groups. Median follow-up was 6 (IQR 2–10) years for WT and 5 (IQR 2–9) years for heterozygous and homozygous *HRC* carriers. However, no statistically significant differences were found between the three groups.

In cohort 2 (ACM, Table 2), approximately half of the patients included were males (WT = 50%, heterozygous = 47%, and homozygous = 55%), while 45% of the WT, 48% of heterozygous, and 47% of homozygous *HRC* carriers were index patients. The median follow-up was 8 [4–15] years for WT and heterozygous carriers and 10 [3–14] years for homozygous *HRC* carriers. For all parameters, no significant differences between groups were found.

Finally, in cohort 3 (DCM, Table 3), the mean age at enrollment was 61 years for all three polymorphism groups. Compared to cohorts 1 and 2, predominantly males were included in the analysis: 71% in WT, 68% for heterozygous, and 79% for homozygous carriers. The mean follow-up time of all three *HRC* groups was approximately 10 years. No statistical differences were found between the groups.

### 2.2. HRC p.(Ser96Ala) Polymorphism in General and Cardiomyopathy Populations

The minor allele frequency (MAF) of the *HRC* p.(Ser96Ala) polymorphism in the control cohorts ranged from 40.3 to 42.2%, as analyzed in different databases. 

In the three patient cohorts, the MAF of the *HRC* p.(Ser96Ala) polymorphism was 40.9% among *PLN* p.(Arg14del) carriers (cohort 1), 43.9% for the overall ACM cohort (cohort 2), and 41.3% for the UK Biobank DCM cohort (cohort 3). In more detail, the Dutch ACM cohort (*n* = 491) showed a MAF of 43.6%, while a MAF of 43.0% was found among US ACM patients (*n* = 337) and 50.9% among the Swiss ACM cohort (*n* = 54). Data regarding the different observed frequencies are summarized in Table 4.

### 2.3. Life-Threatening Ventricular Arrhythmias and HRC Polymorphism

Among *PLN* p.(Arg14del) carriers from cohort 1, life-threatening VA events were observed in 49/294 (16%) of patients with WT p.(Ser96Ala), 54/392 (14%) of heterozygotes, and 16/93 (17%) of p.(Ser96Ala) homozygotes (Table 1). Logistic regression revealed no significant association between p.(Ser96Ala) and life-threatening VA among *PLN* p.(Arg14del) pathogenic variant carriers (Odds Ratio (OR) (95% confidence interval (CI)) 0.792 (0.584–1.066), *p* = 0.128; Table 5). Moreover, in prespecified analyses of subgroups (restricting to index patients or relatives), only a significant association between the p.(Ser96Ala) polymorphism and life-threatening VA in index patients was found (OR (95% CI) = 0.554 (0.300–0.985), *p* = 0.049 *), but after correction for multiple testing, this statistical result was not sustained (Appendix A). Finally, time-to-event analyses revealed that the p.(Ser96Ala) polymorphism was not significantly associated with life-threatening VA when subjected from birth or from enrollment into the registry (Appendix A).

Among ACM patients from cohort 2, life-threatening VA events were observed in 120/283 (42%) WT carriers, 186/375 (50%) heterozygotes, and 77/174 (44%) p.(Ser96Ala) homozygotes. *HRC* status was not statistically associated with life-threatening arrhythmias in this patient group (OR (95% CI) 0.862 (0.576–1.288), *p* = 0.467, Table 6), further supporting the results observed among *PLN* carriers in cohort 1.

We then assessed the risk of events by p.(Ser96Ala) status among patients with DCM using UK Biobank samples from cohort 3. Logistic regression among these DCM cases also revealed no significant association with different arrhythmic parameters, including ventricular tachycardia (VT) (OR (95% CI) 0.848 (0.524–1.171), *p* = 0.210), SCD (OR (95% CI) 1.016 (0.985–1.046), *p* = 0.941), or implantable cardioverter defibrillator (ICD) implantation (OR (95% CI) 0.887 (0.652–1.121), *p* = 0.284, Table 7). Time-to-event analyses similarly yielded insignificant results, with hazard ratios inconsistent with large effects (Appendix A). In addition, there was no evidence of increased risk for all-cause mortality (HR (95% CI) 1.140 (0.943–1.378), *p* = 0.172). 

### 2.4. Heart Failure and HRC Polymorphism

Among *PLN* p.(Arg14del) carriers from cohort 1, HF was observed in 42/257 (16%) WT *HRC* carriers, 53/345 (15%) that were heterozygous, and 15/113 (13%) p.(Ser96Ala) homozygotes (Table 1). No significant association was found between the *HRC* polymorphism and the HF outcome using logistic regression (OR (95% CI) 0.858 (0.615–1.188), *p* = 0.360; Table 5). The prevalence and severity of progressive HF in ACM might be low given a currently debated definition of HF among these cases, particularly in the right dominant subforms of the disease [17]. In addition, overlapping criteria for congestive HF and DCM are frequently used in the research literature (i.e., LVEF < 45%). For these two reasons, no analyses for the risk of HF outcomes were performed for cohorts 2 and 3.

### 2.5. Composite Endpoint in PLN p.(Arg14del) Patients

In *PLN* p.(Arg14del) carriers (cohort 1), a composite endpoint was observed in 67/302 (22%) of the WT *HRC* carriers, 81/404 (20%) of the heterozygous carriers, and 24/142 (17%) for homozygous *HRC* carriers. No effect of the p.(Ser96Ala) polymorphism with this composite endpoint in *PLN* p.(Arg14del) carriers was observed when adjusting for the prespecified covariates (OR (95% CI) 0.842 (0.649–1.089), *p* = 0.193; Table 5).

## 3. Discussion

In this study, we examined the role of the previously identified *HRC* p.(Ser96Ala) polymorphism in relation to major cardiac events in clinical cohorts diagnosed with, or predisposed to, different forms of cardiomyopathy. Most carriers develop symptoms between their third or fifth decade of life, suggesting an incomplete penetrance. In addition, a wide range of symptoms among patients carrying the same genetic variant can be found [18]. Therefore, improvement in proper risk stratification is needed in these patients. We established that the *HRC* p.(Ser96Ala) polymorphism is frequent (40–42%) in the general population, with highly similar frequencies among the different studied patient cohorts. Importantly, we found that the p.(Ser96Ala) polymorphism did not significantly contribute to the risk of major cardiac events among *PLN* p.(Arg14del) carriers, ACM patients, or DCM patients, contrasting with previous reports.

Recently, studies have started to explore the role of common genetic variants in modifying the risk of cardiomyopathies. Genome-wide association studies for DCM have identified several common genetic variants associated with disease risk [19]. Notably, common genetic variants have been shown to modify the penetrance and expressivity of HCM in patients with rare variants [20]. Given the marked incomplete penetrance and variable expressivity of rare variants—and lack of known pathogenic variants in many index patients with ACM and DCM—common variants may similarly contribute to expressivity in these cardiomyopathies. This is especially relevant as the prediction of major adverse events, including HF and VA, remains exceedingly challenging. 

In the present study, we assessed the risk of HF and VA among a broad range of patient cohorts, namely, *PLN* p.Arg14del carriers, ACM index patients and family members, and DCM patients. This approach was chosen given that the phenotypic outcomes of these entities share overlapping features, among which there is a significant risk of life-threatening VA. Classically, ACM was described as arrhythmogenic right ventricular cardiomyopathy (ARVC) with sole involvement of the right ventricle; however, biventricular and left dominant forms are now increasingly recognized [5]. Conversely, a decline in right ventricular function is a predictor of worse outcomes in DCM patients [21]. Carriers of *PLN* p.(Arg14del) can be diagnosed with either of these cardiomyopathies, reflecting the clinical and genetic overlap between these disease entities [10]. Finally, in both ACM and DCM, disturbance of Ca^2+^ handling is known to drive the development of life-threatening arrhythmias and HF [22,23].

Given this apparent functional mechanism, the p.(Ser96Ala) polymorphism could potentially influence Ca^2+^ regulation and thereby contribute to VA or deterioration of Ca^2+^ handling. Indeed, a marked effect was described in an initial study of 123 idiopathic DCM patients—where p.(Ser96Ala) homozygotes had an over four-fold risk of major arrhythmic events [16]. In contrast, we were unable to confirm a role for the *HRC* p.(Ser96Ala) polymorphism in stratifying the risk of major events among three larger cohorts of patients with (predisposition to) cardiomyopathy. Furthermore, directionally inconsistent results were found in our study compared to the study of Arvanitis et al., as the p.(Ser96Ala) polymorphism seemed to be protective for major cardiac events [16]. We note that several of our cohorts included individuals with preclinical disease or genetic predisposition only, which contrasts with the established DCM patients analyzed in the work of Arvanitis et al.; however, we performed several sensitivity analyses, including index patients only, which generally showed consistent effects. In addition, the one significant association between the index and life-threatening VA, even if not significant after correction for multiple tests, was directionally inconsistent compared to the aforementioned study. Furthermore, it is possible that our study remains underpowered to detect a small effect of p.(Ser96Ala) on major events. Nevertheless, the 95% CI of our estimates—in all cohorts and analyses—excludes a large effect, especially one as large as described by Arvanitis et al.

## 4. Materials and Methods

### 4.1. Study Design and Patient Selection

In this study, individuals from four different patient and control cohorts were included:

Cohort 1 included individuals of Dutch ancestry enrolled in the Dutch ACM registry in whom the pathogenic p.(Arg14del) variant in *PLN* was identified (accessed on 7 March 2023). Within this database, we selected *PLN* p.(Arg14del) carriers, and this will be referred to as the *PLN* registry. This registry includes data from individuals obtained from three Dutch university medical centers (Groningen, Amsterdam, and Utrecht) [24]. The study cohort consisted of both index patients (i.e., first affected family member tested positive for the disease-associated genetic variant, mostly because of suspected inherited cardiac disease) and relatives after genetic cascade testing.

Cohort 2 included individuals enrolled in the ACM registries from the Netherlands [24], Switzerland, and The United States (USA, data accessed on 8 March 2023). Index and relatives with a definitive ARVC diagnosis based on the 2010 modified Task Force Criteria (TFC) and relatives with a pathogenic variant in an ARVC-related gene (who do not meet TFC) or gene elusive were included.

Cohort 3 included DCM patients identified within the UK Biobank cohort. Patients with a clinical DCM diagnosis were identified using International Classification of Diseases (ICD-10 code I42.0) [25]. 

Control cohort included individuals that belong to the “general population”. Population data consisted of three publicly available databases: (1) Genome Aggregation Database (gnomAD v2.1.1, https://gnomad.broadinstitute.org/, accessed on 23 March 2023), (2) 998 individuals from The Genome of the Netherlands (GoNL, accessed on 25 May 2023) [26], and (3) participants of the UK Biobank (accessed on 26 April 2023). The UK Biobank is a large population-based prospective study that included 500,000 UK participants between 40 and 69 years [27]. Additionally, 31,400 individuals who were referred to the Department of Genetics of the UMC Utrecht, the Netherlands (inclusion between 2017 and 2023, data accessed on 5 May 2023) were evaluated irrespective of diagnosis. In this group, a whole exome sequencing (WES)-based genetic test was available as part of regular clinical care.

For all cohorts, only individuals with European (-American) ancestry were selected for further analysis. Written informed consent was provided by all participating patients. Furthermore, all data were fully anonymized before data could be accessed. This study was conducted according to the Declaration of Helsinki. Study protocol was approved by UMC Utrecht (Biobank number 12–387). Use of UK Biobank data was performed under application number 17,488 and was approved by the local Massachusetts General Hospital Institutional Review Board.

### 4.2. Genetic Data Extraction

Cohort 1 and 2: DNA samples of *PLN* p.(Arg14del) carriers, ACM patients, and preclinical gene variant carriers enrolled in the ACM registry were genotyped on the Illumina Global Screening Array (-24 v3.0 BeadChip, San Diego, CA, USA) at the Human Genomics Facility (HuGe-F), Erasmus Medical Center, Rotterdam, the Netherlands and at the Genetic Resources Core Facility (GRCF), at John Hopkins University, Baltimore, the USA. The p.(Ser96Ala) polymorphism (T/G) (rs3745297) in *HRC* was extracted on imputed data after standard sample QC (INFO score > 0.99; removal of samples was performed for those who revoked consent, had a mismatch between genetically predicted and self-reported sex, were outliers for heterozygosity or missingness, had putative sex chromosome aneuploidy, or were ancestral outliers in a Principal Component (PC) Analysis) using Plink v1.9 and v2.0. The *HRC* polymorphism was categorized as either WT (TT), heterozygous (TG), or homozygous (GG).

Cohort 3: UK Biobank participants underwent dense genotyping using the UK Biobank Axiom Array or the Affymetrix UK BiLEVE Axiom Array (Thermo Fisher Scientific, Waltham, MA, USA), as described by Bycroft et al. [28]. We used the version 3 imputed data, where the p.(Ser96Ala) polymorphism attained near-perfect imputation accuracy (INFO > 0.99). Standard sample QC was performed as described before.

Control cohort: the *HRC* p.(Ser96Ala) polymorphism was extracted from gnomAD (WES data used) [29]; GoNL (whole-genome sequencing (WGS) data used) [26]; the UK Biobank (version 3 imputed data; INFO score > 0.99; sample QC described above) [28]; and in patients in whom WES was performed as part of regular clinical care, irrespective of diagnosis, in UMC Utrecht, Utrecht, The Netherlands between 2017 and 2023.

### 4.3. Clinical Outcomes

The first primary outcome was defined as the first occurrence of a life-threatening VA event, including sustained VT or ventricular fibrillation (VF), appropriate ICD therapy, and (aborted) SCD. The second primary outcome was defined as the first occurrence of an HF-related event. This included hospitalization for HF-related complaints, implantation of a left ventricular assistance device (LVAD), heart transplantation, or HF-related death. A secondary outcome was a composite of both life-threatening VA and/or HF-related events. For cohorts 1–2, clinical information from the first cardiac evaluation and follow-up visits was retrospectively extracted from the ACM and *PLN* registries, as described before [24]. Age at enrollment in the registry was defined by presenting with clinical presentation due to cardiomyopathy-related symptoms, SCD, or family screening [24]. 

For the UK Biobank DCM cohort (cohort 3), participants attended an entry assessment at centers across the UK to provide baseline characteristics. Follow-up data were collected via hospital event data and death registry linkage; main outcomes (VT, SCD, and ICD implantation) were defined using ICD codes, as previously described [25]. The latest update of health record linkage data was performed in October 2020. 

### 4.4. Statistical Analysis

#### 4.4.1. Registry Cohorts

Outcomes and covariates of interest were collected and presented as descriptive statistics for each registry cohort; categorical variables were analyzed using Chi-Square tests, and continuous variables were analyzed using Kruskal–Wallis tests. In each dataset, logistic regression was performed with life-threatening VA, HF, or both (composite outcome) modeled as outcome; the *HRC* p.(Ser96Ala) was modeled as a predictor, adjusting for covariates of sex, age at enrollment (in the registry cohorts), and the first 12 PCs of ancestry. Logistic regression analyses were repeated within subgroups, i.e., restricting to index cases or relatives. Additionally, we performed analyses using only incident events modeled in a Cox proportional hazard regression (time-to-event). Cox models were adjusted for the same covariates as in the logistic regression analysis and right-censored at last follow-up at the cardiology outpatient clinic or death if the primary outcome had not occurred. Start of incident time was modeled in two ways, namely, (1) from birth (which effectively is true since the genetic variant is already present at birth) and (2) from enrollment in the ACM or *PLN* registry. In addition to the aforementioned outcomes, mortality was also assessed in time-to-event analyses. Differences were considered statistically significant when *p* < 0.05. Statistical analyses were performed with R version 4.0.3, using the “survival” and “survminer” packages [30,31]. 

#### 4.4.2. UK Biobank DCM Cohort

In the UK Biobank DCM cohort, logistic regression was used, pooling incident and prevalent events to maximize case numbers and power. Outcomes included VA, SCD, and ICD implantation. Models were adjusted for age, sex, genotyping array, and the first 12 PCs of ancestry. Because logistic regression may not correctly model temporality and competing risks, we also performed analyses using only incident events modeled in a Cox proportional hazard regression, using function coxph() from R package "survival" [30]. Cox models were adjusting for the same covariates and right-censored at death to account for competing risks of death. Start of incident time was modeled in two ways, namely, (1) from enrollment in UK Biobank and (2) from DCM diagnosis. In addition to the aforementioned outcomes, mortality was also assessed in time-to-event analyses. Differences were considered statistically significant when *p* < 0.05.

## 5. Conclusions

In this study, we found that the p.(Ser96Ala) polymorphism is common in the general population and cardiomyopathy-affected patients. Furthermore, there is a lack of evidence for the role of the p.(Ser96Ala) *HRC* polymorphism in modifying the risk of major cardiac events among cardiomyopathy patients. Our data indicate that any possible effect is, at most, small, limiting its use as a sole predictor of severity among ACM and DCM patients. Further research is required to identify bona fide predictors for stratification of cardiomyopathy patients and their risk for life-threatening outcomes.

### Limitations

Despite the unique (and rare) composition of our included cohorts, this study could be subjected to several limitations. A larger part of the *PLN* p.(Arg14del) pathogenic variant carriers and ACM patients are classified as still being in the preclinical phase. Therefore, clinical events might be low compared to a cohort that includes only diagnosed patients, such as the DCM patients from the UK Biobank. In addition, lifestyle factors such as diet, environment, and exercise might influence and accelerate disease manifestation and events. However, these data were not available when we performed our analyses. In addition, some clinical, MRI, or echocardiographic parameters have not been reliably collected (e.g., medication use) or were not available. However, we have the impression that the unavailability of these data does not hamper the major conclusions and genetic observations of this study. When estimating a population MAF, most of the studies make use of a controlled setting in which cases and controls are matched. However, whether this group represents the population frequencies might not be taken into account [32]. Therefore, extrapolation of the observed allele frequencies to the general population can be inadequate. Finally, we note that our study was largely focused on individuals of European ancestry, potentially limiting the generalizability to other ancestry groups. As illustrated in a single study on Japanese paroxysmal atrial fibrillation patients, a MAF of 26% was reported, which was similar to population data of East Asian ancestry found in gnomAD [33].

## Figures and Tables

**Figure 1 ijms-24-15931-f001:**
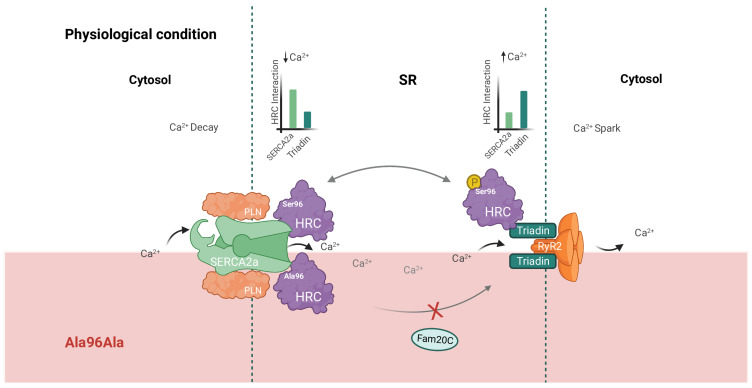
Schematic overview of the interactions of HRC protein in the SR. At low [Ca^2+^] in the SR, HRC protein interacts with SERCA2a. When [Ca^2+^] rises, HRC is phosphorylated by Fam20C at Ser96 position, which induces interaction with triadin to modulate RyR2 function. When the serine residue is altered for an alanine residue (Ala96Ala variant), phosphorylation is not performed, and therefore affinity for SERCA2a and triadin remains unaffected. HRC; histidine-rich calcium-binding protein, SR; sarcoplasmic reticulum, Ca^2+^; calcium, SERCA; sarcoplasmic/endoplasmic reticulum Ca^2+^-ATPase, PLN; phospholamban, RyR2; ryanodine receptor 2, Ser; serine, Ala; alanine, Fam20C; family with sequence similarity 20C.

**Figure 2 ijms-24-15931-f002:**
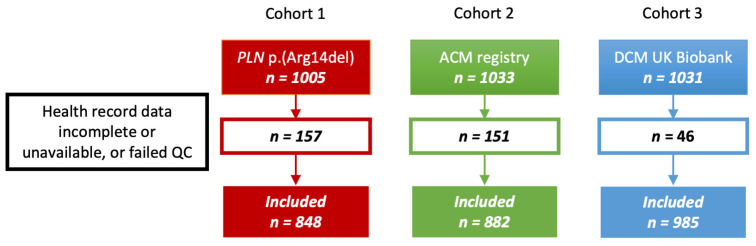
Flowchart inclusion of carriers of the *PLN* p.(Arg14del) pathogenic variant (cohort 1); patients diagnosed with, or predisposed to, ACM (cohort 2); and DCM patients (cohort 3). *PLN*; phospholamban, ACM; arrhythmogenic cardiomyopathy, DCM; dilated cardiomyopathy, UK; United Kingdom, QC: quality control.

**Table 1 ijms-24-15931-t001:** Patient characteristics of the 848 *PLN* p.(Arg14del) carriers included in this study. Patients were divided into wild type for *HRC* p.(Ser96Ser) variant carriers (N = 302), heterozygous for *HRC* p.(Ser96Ala) (N = 404), and homozygous carriers p.(Ala96Ala, N = 142). Data are depicted as median (interquartile range) or *n*/N (%).

PLN p.(Arg14del) N *= 848*	Wild Type (TT) (N = 302, 36%)	Heterozygous (TG) (N = 404, 48%)	Homozygous (GG) (N = 142, 16%)
**Age (years)**	51 [37–63]	52 [36–65]	49 [32–62]
**Male sex**	131/302 (43%)	182/404 (45%)	64/142 (45%)
**Index patient**	61/302 (20%)	78/404 (19%)	23/142 (16%)
**Diagnosis**			
DCM	42/195 (22%)	61/257 (24%)	21/100 (21%)
ACM	21/184 (11%)	27/254 (11%)	6/96 (6%)
**ICD implantation**	90/267 (34%)	115/356 (32%)	43/120 (36%)
**Continuous rhythm monitoring**			
≥500 PVCs	61/190 (32%)	93/250 (37%)	28/88 (32%)
**Imaging**			
LVEF (%)	51 [41–56]	52 [41–57]	50 [43–55]
**Life-threatening arrhythmias (MVA)**	49/294 (16%)	54/392 (14%)	16/139 (17%)
(Aborted) SCD	4 (1%)	10 (3%)	2 (1%)
Sustained VT	36 (12%)	43 (11%)	15 (11%)
Appropriate ICD shock	15 (5%)	20 (5%)	5 (4%)
**HF-related events**	42/257 (16%)	53/345 (15%)	15/113 (13%)
Hospitalization for HF	37 (14%)	37 (11%)	13 (11%)
HTx or LVAD	18 (7%)	28 (8%)	7 (6%)
HF death	18 (7%)	17 (4%)	3 (3%)
**Composite of MVA and/or HF events**	67/302 (22%)	81/404 (20%)	24/142 (17%)
**Follow-up clinical evaluation (years)**	6 [2–10]	5 [2–9]	5 [2–9]

PLN; phospholamban, DCM; dilated cardiomyopathy, ACM; arrhythmogenic cardiomyopathy, ICD; implantable cardioverter defibrillator, PVC; premature ventricular contraction, LVEF; left ventricular ejection fraction, MVA; major ventricular arrhythmias, SCD; sudden cardiac death, VT; ventricular tachycardia, HF; heart failure, HTx; heart transplantation, LVAD; left ventricular assistance device.

**Table 2 ijms-24-15931-t002:** Patient characteristics of the 882 ACM patients included in this study. Patients were divided into wild type for *HRC* p.(Ser96Ser) variant carriers (N = 288), heterozygous for *HRC* (p.(Ser96Ala), N = 413), and homozygous carriers p.(Ala96Ala), N = 181). Data are depicted as median (interquartile range) or *n*/N (%).

ACM (N = 882)	Wild Type (TT) (N = 288, 33%)	Heterozygous (TG) (N = 413, 47%)	Homozygous (GG) (N = 181, 20%)
**Age (years)**	47 [33–60]	47 [33–57]	48 [33–58]
**Male sex**	144/288 (50%)	196/413 (47%)	99/181 (55%)
**Index patient**	129/287 (45%)	195/403 (48%)	87/184 (47%)
**Diagnosis**			
ACM	178/200 (89%)	245/272 (90%)	108/117 (92%)
**Life-threatening** **arrhythmias (MVA)**	120/283 (42%)	186/375 (50%)	77/174 (44%)
**HF-related events**	19/142 (13%)	14/171 (8%)	13/92 (14%)
HTx or LVAD	11 (8%)	15 (9%)	5 (5%)
**Follow-up clinical evaluation (years)**	8 [4–15]	8 [4–15]	10 [3–14]

ACM; arrhythmogenic cardiomyopathy, HF; heart failure, HTx; heart transplantation, LVAD; left ventricular assistance device.

**Table 3 ijms-24-15931-t003:** Patient characteristics of the 985 DCM patients included in this study. Patients were divided into wild type for *HRC* p.(Ser96Ser) variant carriers (N = 316), heterozygous for *HRC* (p.(Ser96Ala), N = 524), and homozygous carriers (p.Ala96Ala, N = 145). Data are depicted as mean ± standard deviation (SD) or *n* (%).

DCM (N = 985)	Wild Type (TT) (N = 316, 32%)	Heterozygous (TG) (N = 524, 53%)	Homozygous (GG) (N = 145, 15%)
**Enrollment age in years**	60.6 ± 7.1	60.5 ± 6.8	60.5 ± 6.6
**Male sex**	225 (71%)	357 (68%)	114 (79%)
**VT**	62 (20%)	80 (15%)	24 (17%)
**(Aborted) SCD**	18 (6%)	30 (6%)	8 (6%)
**VT or SCD**	72 (23%)	91 (17%)	27 (19%)
**ICD implantation**	90 (28%)	139 (27%)	34 (23%)
**Mortality**	72 (23%)	131 (25%)	45 (31%)
**Biobank follow-up time in years**	10.2 ± 2.3	10.3 ± 2.6	9.8 ± 3.2

DCM; dilated cardiomyopathy, VT; ventricular tachycardia, SCD; sudden cardiac death, ICD; implantable cardioverter defibrillator.

**Table 4 ijms-24-15931-t004:** Minor allele frequencies of the *HRC* polymorphism in general and registry cohorts.

	Frequencies *HRC* Polymorphism
**General Population**	
gnomAD, European (non-Finish)	41.7%
GoNL	40.5%
WES, UMC Utrecht	40.3%
UK Biobank	42.2%
***PLN*** **Registry**	40.9%
**ACM Registry**	43.9%
Dutch	43.6%
USA	43.0%
Swiss	50.9%
**DCM Cohort, UK Biobank**	
British	41.3%

HRC; histidine-rich calcium-binding protein, gnomAD; genome aggregation database, GoNL: genome of the Netherlands, WES; whole-exome sequencing, UMC; university medical center, UK; United Kingdom, PLN; phospholamban, ACM; arrhythmogenic cardiomyopathy, USA; United States, DCM; dilated cardiomyopathy.

**Table 5 ijms-24-15931-t005:** Logistic regression analyses in *PLN* p.(Arg14del) carriers were modeled with *HRC* p.(Ser96Ala) as predictor. Analyses were adjusted for sex, age at enrollment, and principal components.

*PLN* p.(Arg14del)	Odds Ratio	95% CI	*p*-Value
**Life-threatening VA event**	0.792	0.584–1.066	0.128
**HF event**	0.858	0.615–1.188	0.360
**Composite**	0.842	0.649–1.089	0.193

PLN; phospholamban, CI; confidence interval, VA; ventricular arrhythmia, HF; heart failure.

**Table 6 ijms-24-15931-t006:** Logistic regression analysis in ACM patients was modeled with *HRC* p.(Ser96Ala) as predictor. Analyses were adjusted for sex, age at enrollment, and principal components.

ACM	Odds Ratio	95% CI	*p*-Value
**Life-threatening VA event**	0.862	0.576–1.288	0.467

ACM; arrhythmogenic cardiomyopathy, CI; confidence interval, VA; ventricular arrhythmia.

**Table 7 ijms-24-15931-t007:** Logistic regression analyses in DCM patients were modeled with *HRC* p.(Ser96Ala) as predictor. Analyses were adjusted for sex, age at enrollment, genotype array, and principal components.

DCM	Odds Ratio	95% CI	*p*-Value
**VT**	0.848	0.524–1.171	0.210
**SCD**	1.016	0.985–1.046	0.941
**ICD** **implantation**	0.887	0.652–1.121	0.284

DCM; dilated cardiomyopathy, CI; confidence interval, VT; ventricular tachycardia, SCD; sudden cardiac death, ICD; implantable cardioverter defibrillator.

## Data Availability

The data presented in this study are available on request from the corresponding author.

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
