# Peer review of "Lack of Evidence for the Role of the p.(Ser96Ala) Polymorphism in Histidine-Rich Calcium Binding Protein as a Secondary Hit in Cardiomyopathies"

_ijms, 2023, doi:10.3390/ijms242115931_

Round 1
Reviewer 1 Report
Comments and Suggestions for Authors
In the article ‘Lack of Evidence for the Role of the Ser96Ala Polymorphism in Histidine-Rich Calcium Binding Protein as a Secondary Hit in Cardiomyopathies’ submitted by van der Voorn and coworkers to the International Journal of Molecular Sciences, the authors performed a genetic analysis about a specific HRC polymorphism.
The topic is interesting. The data of this study are showing that the hypothesis of HRC involvement in cardiomyopathies is not likely. However, I have some points which should be corrected by the authors:
1.) Phospholamban is a protein name and should not be written in Italics. The gene name is PLN, which should be written in Italics.
2.) Please add the cDNA nomenclature, when you introduce a specific variant the first time.
3.) Please add OMIM identifiers for ACM and DCM.
4.) In general, it would be helpful, if you introduce shortly the genetic background of ACM or you can reference a good review article to this topic: Insights Into Genetics and Pathophysiology of Arrhythmogenic Cardiomyopathy. Gerull B, Brodehl A. 2021
5.) The first sentence of the introduction is misleading. In the majority of cases an autosomal dominant inheritance can be found. However, there are several reports indicating a recessive inheritance. For example, the homozygous DSC2 mutation c.1913_1916delAGAA, p.Q638LfsX647hom was described in an ACM patient. Similarly, also hemi- and homozygous double mutations in DSG2, encoding desmoglein-2, have been recently described. Please add this information for recessive inheritance in specific cases to the introduction including relevant citations.
In summary, I suggest a minor revision for this manuscript. It is in my view also important to publish negative results!
Comments on the Quality of English LanguageSome corrections might be necessary by a native speaking editor.
Author Response
We would like to thank the reviewer for his/her efforts and constructive advice regarding our manuscript. Below, we have addressed the reviewer’s comments in a point-wise fashion and revised the manuscript accordingly (track changed). We believe by implementing the obtained suggestions the manuscript has been significantly improved.
- Phospholamban is a protein name and should not be written in Italics. The gene name is PLN, which should be written in Italics.
We agree and have changed it throughout the manuscript. Phospholamban is now no longer written in italics.
2.) Please add the cDNA nomenclature, when you introduce a specific variant the first time.
The cDNA nomenclature has been added for specific variants when introduced for the first time and can be found in lines 35 and 100.
3.) Please add OMIM identifiers for ACM and DCM.
Unfortunately, we were not able to include OMIM identifiers as in our study we are not referring to variants in one gene in particular. The OMIM identifiers are linked to both disease and gene/genomic location. There could be potentially one exception in cohort 1: the PLN p.(Arg14del) pathogenic variant. However, there is no OMIM identifier for (likely) pathogenic variants in PLN related to ACM. We included the OMIM identifier for PLN p.(Arg14del) carriers with a DCM phenotype (609909) As described in in the method section, we selected for diagnosis and therefore also included gene elusive patients in our ACM and DCM cohort, see line 79.
4.) In general, it would be helpful, if you introduce shortly the genetic background of ACM or you can reference a good review article to this topic: Insights Into Genetics and Pathophysiology of Arrhythmogenic Cardiomyopathy. Gerull B, Brodehl A. 2021
We thank the reviewer for this suggestion, we added this specific reference in the introduction of our manuscript (line 76).
5.) The first sentence of the introduction is misleading. In the majority of cases an autosomal dominant inheritance can be found. However, there are several reports indicating a recessive inheritance. For example, the homozygous DSC2 mutation c.1913_1916delAGAA, p.Q638LfsX647hom was described in an ACM patient. Similarly, also hemi- and homozygous double mutations in DSG2, encoding desmoglein-2, have been recently described. Please add this information for recessive inheritance in specific cases to the introduction including relevant citations.
We do agree with the reviewer that the first sentence of the introduction could be confusing and therefore we rewrote this sentence so that it now reads: ”Inherited cardiomyopathies represent relatively rare but morbid diseases and may lead to severe heart failure (HF), life-threatening ventricular arrhythmias (VA), and even sudden cardiac death (SCD) at young age. Prevalence of inherited cardiomyopathies are estimated 1/250 to 1/500 for hypertrophic cardiomyopathy (HCM), 1/250 for dilated cardiomyopathy (DCM) and 1/5000 for arrhythmogenic cardiomyopathy (ACM).”

Reviewer 2 Report
Comments and Suggestions for Authors
The authors investigated whether Ser96Ala polymorphism in 2 Histidine-Rich Calcium Binding Protein is predictors and biomarker for cardiomyopathy. They found that allele frequency of the Ser96Ala polymorphism was similar between the general population and the different cardiomyopathy cohorts. Importantly the Ser96Ala polymorphism did not contribute to risk of major cardiac events. Although it is a negative result, the study provides new knowledge about the prediction of cardiomyopathy. The data is convincing and trustable.
For the first paragraph of Introduction, it is suggested that the authors moved the last sentence behind the first sentence. Then it will not interrupt the introduction.
In Results 2.1, the authors should define the characteristic of general population. It is suggested that the authors remove the fourth paragraph of Discussion and relevant Figure 2. It extends the research field of this study and increases the complexity of discussion. If the authors want to keep them, it is better to present them in the Introduction part.All the references are appropriate in this manuscript.
Author Response
We would like to thank the reviewer for his/her efforts and constructive advice regarding our manuscript. Below, we have addressed the reviewer’s comments in a point-wise fashion and revised the manuscript accordingly (track changed). We believe by implementing the obtained suggestions the manuscript has been significantly improved.
1. For the first paragraph of Introduction, it is suggested that the authors moved the last sentence behind the first sentence. Then it will not interrupt the introduction.
As suggested by the reviewer, we moved the last sentence of the first paragraph to the first sentences, so that it now reads: “Inherited cardiomyopathies represent relatively rare but morbid diseases and may lead to severe heart failure (HF), life-threatening ventricular arrhythmias (VA), and even sudden cardiac death (SCD) at young age. Prevalence of inherited cardiomyopathies are estimated 1/250 to 1/500 for hypertrophic cardiomyopathy (HCM), 1/250 for dilated cardiomyopathy (DCM) and 1/5000 for arrhythmogenic cardiomyopathy (ACM)."
2. In Results 2.1, the authors should define the characteristic of general population.
We used frequency data of publicly available cohorts, which were defined as ‘the general population’ (control cohort). We only selected cohorts with a European(-American) ancestry in order that the diseased and control cohort have the same genetic background, but we did not select participants based on e.g. sex or age. Phenotypic characteristics of these population data is very limited available. We don’t think that defining additional characteristics of the general population will improve our study.
3. It is suggested that the authors remove the fourth paragraph of Discussion and relevant Figure 2. It extends the research field of this study and increases the complexity of discussion. If the authors want to keep them, it is better to present them in the Introduction part.
Like suggested by the reviewer we have moved the fourth paragraph of the Discussion and Figure 2 to the introduction. This also means we renumbered the figures. We did not remove our explanation of the HRC function in the SR, because we think it is illustrative for the reader to understand why HRC is an interesting target for this study, see lines 90-99.

Reviewer 3 Report
Comments and Suggestions for Authors
The authors aimed to investigate the contribution of HRC Ser96Ala polymorphism in major disease manifestations of different inherited cardiomyopathy cohorts.
The paper is well written, however some additional points should be addressed
1) What is the percentage of patients who underwent cardiac magnetic resonance imaging? From these how many were assessed using Late gadolinium enhancement (LGE) please report this in results and statistical analysis.
2) Medication at enrolment should be reported (mainly, Betablockers, Valsartan/Sacubitril, Betablockers, Amiodaron)
3) What is the percentage of patients with left bundle branch block (LBBB) in the different cohorts? Why was this parameter not included in study?
4) Important echocardiographic parameters should be included as LVEDD and LVESD.
5) NYHA classes should be mentioned in the tables.
Comments on the Quality of English LanguageMinor editing is needed
Author Response
The paper is well written, however some additional points should be addressed
1) What is the percentage of patients who underwent cardiac magnetic resonance imaging? From these how many were assessed using Late gadolinium enhancement (LGE) please report this in results and statistical analysis.
2) Medication at enrolment should be reported (mainly, Betablockers, Valsartan/Sacubitril, Betablockers, Amiodaron)
3) What is the percentage of patients with left bundle branch block (LBBB) in the different cohorts? Why was this parameter not included in study?
4) Important echocardiographic parameters should be included as LVEDD and LVESD.
5) NYHA classes should be mentioned in the tables.
Point 1-5: We would like to thank the reviewer for his/her efforts and constructive advice regarding our manuscript. We fully understand the rationale behind, and the relevance of the suggestions made by the reviewer. Unfortunately, it is not feasible to provide the requested data within the short turn-around time of this rebuttal. More importantly, some data have not been reliably collected (medication use, LBBB), and as such are not available. We also have the impression that the unavailability of these data does not really hamper the major conclusions and genetic observations of this study, namely that there is no real difference in the observed presence of the HRC polymorphism in the different populations emphasizing that Ser96Ala is not as relevant as initially observed.
But again, we acknowledge this limitation and have added this as a limitation to our study (discussion lines 681-685).

Reviewer 4 Report
Comments and Suggestions for Authors
Dear Authors,
the topic is relevant and interesting. The concerns raised are primarily for better readability and understanding of the data.
Abstract: generally, it helps if there is some sectioning in the abstract which allows a fast a clear capture of background, aim, conclusion etc. it is kind of confusing to present cohort 1 after cohorts 2-4. please present also numbers e.g., for allele frequency.
Introduction: “relatively rare”, please state with epidemiologic numbers what you mean by that. Referenced work should not be a review article, but rather an original study (e.g., lines 59ff). Is there an observation of Ser96Ala polymorphism in the general population/other cohorts (lines 83ff)?
Results: It again confuses that cohort 1 is left out; maybe it would help to name the control group (general population) as cohort 4. Line 104: “Tables 1, 2 and 3 …” please add the specific group to the sentence: “Tables 1-3 show the specific groups, i.e., PLN p.(Arg14del), ACM and DCM, respectively.” Otherwise, orientation is hard. Tables: the total number should appear somewhere, best if below the group name. A measure of significance between the groups is missing. Table 1: “Diagnosis” and “Imaging” are on the left; it is hard to see why. Tables 5-7: in the introduction you mention that inherited cardiomyopathies may lead to HF, VA and SCD. why don’t you show those parameters in these tables? If possible, I would suggest to include all cohorts in one table with the parameters on the left/down and the cohorts on the top. you may give OR (95%CI) and p-value for each. The estimate itself may be of limited use here.
Discussion: Lines 215ff incomplete penetrance and ..expression, please give numbers to better explain to the readers. And discuss disease polymorphism frequency in the literature, see above. Lines 243-252 incl Fig 2 should belong to the introduction. The dialogue with Arvanitis et al. is obviously important, but there are also other data.. please mention.
Materials: Clinical outcomes (primary and secondary endpoints) are important and should be presented within the tables, or better the table, as mentioned above.
Limitations: ancestry groups: are other data/literature similar?
Comments on the Quality of English LanguageIn most part fine, minor typos
Author Response
We would like to thank the reviewer for his/her efforts and constructive advice regarding our manuscript. Below, we have addressed the reviewer’s comments in a point-wise fashion and revised the manuscript accordingly (track changed). We believe by implementing the obtained suggestions the manuscript has been significantly improved.
1. Abstract: generally, it helps if there is some sectioning in the abstract which allows a fast a clear capture of background, aim, conclusion etc.
We fully acknowledge this suggestion, however we have followed the policy of this journal which described to not add captures like background, aim, conclusion etc.
2. it is kind of confusing to present cohort 1 after cohorts 2-4. please present also numbers e.g., for allele frequency.
We understand the confusion and therefore we renamed the patient cohorts as cohort 1: PLN cohort, cohort 2; ACM cohort and cohort 3: DCM. The general population is now addressed as control cohort. In addition, we added the allele frequencies to the abstract.
3. Introduction: “relatively rare”, please state with epidemiologic numbers what you mean by that.
We added epidemiologic numbers of different inherited cardiomyopathies, so now it reads: “Inherited cardiomyopathies represent relatively rare but morbid diseases. Prevalence of inherited cardiomyopathies are estimated 1/250 to 1/500 for hypertrophic cardiomyopathy (HCM), 1/250 for dilated cardiomyopathy (DCM) and 1/5000 for arrhythmogenic cardiomyopathy (ACM)."
4. Referenced work should not be a review article, but rather an original study (e.g., lines 59ff).
We would like to thank the reviewer that you have brought this to our attention. We changed review articles to original studies when possible.
5. Is there an observation of Ser96Ala polymorphism in the general population/other cohorts (lines 83ff)?
To the best of our knowledge, no other studies on the HRC p.(Ser96Ala) allele frequencies in individuals with an identical ethnic background have been published. However, our study is for the first time a description of the observations of the p.Ser96Ala polymorphism in the general population. We did found a Japanese cohort, however this population has a different genetic background compared to patients and control cohorts in our study. Therefore, we choose to not include that study in our manuscript but added it to the limitations, see lines 690-695.
6. Results: It again confuses that cohort 1 is left out; maybe it would help to name the control group (general population) as cohort 4.
We renamed the cohorts and general population. We would like to refer to a previous comment on this topic: the patient cohorts referred to as cohort 1: PLN cohort, cohort 2; ACM cohort and cohort 3: DCM. The general population is now addressed as control cohort.
7. Line 104: “Tables 1, 2 and 3 …” please add the specific group to the sentence: “Tables 1-3 show the specific groups, i.e., PLN p.(Arg14del), ACM and DCM, respectively.” Otherwise, orientation is hard.
We added a specification of the tables to the sentence (lines 186-187) so that it now reads: “Tables 1, 2 and 3 summarize the clinical characteristics of the specific patient cohorts PLN p.(Arg14del), ACM and DCM respectively, dividing the cohorts into wild-type (TT), heterozygous (TG) or homozygous (GG) for the HRC polymorphism.”
8. Tables: the total number should appear somewhere, best if below the group name.
This is a relevant suggestion. We have added the total patient number in tables 1-3.
9. A measure of significance between the groups is missing.
No statistical significance was found between the groups. We added a sentence per cohort to emphasize that no difference was found. See lines 195-196 for cohort 1, “However, no statistically significant differences were found between the three groups”. See line 230 for cohort 2, “For all parameters, no significant differences between groups were found.” And see lines 240-241 for cohort 3, “No statical differences were found between the groups.”
10. Table 1: “Diagnosis” and “Imaging” are on the left; it is hard to see why.
We understand the confusion, therefore we changed the layout of Table 1 and Table 2. We aligned all characteristics to the left side of the table. We used headings like “Diagnosis”, “Continuous rhythm monitoring”, “Imaging”, Life-threatening arrhythmias (MVA)” and “HF-related events” to show where these characteristics were built from, i.e., ACM and DCM belong to the heading of “Diagnosis”.
11. Tables 5-7: in the introduction you mention that inherited cardiomyopathies may lead to HF, VA and SCD. why don’t you show those parameters in these tables? If possible, I would suggest to include all cohorts in one table with the parameters on the left/down and the cohorts on the top. you may give OR (95%CI) and p-value for each. The estimate itself may be of limited use here.
We thank the reviewer for this suggestion, however as parameters extracted from three different registries were not entirely uniform, we chose to show outcomes in multiple separate tables. We changed the terminology in our table to a uniform format whenever possible, because all three parameters (HF, VA, SCD) mentioned were described in the tables. In case a certain parameter was not used in our analyses, we explained the reason for this in our manuscript. For example, why no analyses for HF outcomes were performed for cohorts 2 and 3 (lines 395-399).
12. Discussion: Lines 215ff incomplete penetrance and expression, please give numbers to better explain to the readers.
We further elaborated on incomplete penetrance and expression in the discussion (lines 410-413). So that it now reads “Most carriers develop symptoms between their third or fifth decade of life, suggesting an incomplete penetrance. In addition, a wide range of symptoms among patients carrying the same genetic variant can be found.”
13. And discuss disease polymorphism frequency in the literature, see above.
See previous comment on this topic.
14. Lines 243-252 incl Fig 2 should belong to the introduction.
We fully agree with the reviewer and we moved these lines and Fig 2 to the introduction (lines 90-99).
15. The dialogue with Arvanitis et al. is obviously important, but there are also other data, please mention.
To the best of our knowledge only the group of Arvanitis et al. published several papers on this topic. We also found a single study on a paroxysmal atrial fibrillation in a population with a different ethnic background (Asian). We therefore choose not to include this study, but we added this point to the limitations see lines 690-695.
16. Materials: Clinical outcomes (primary and secondary endpoints) are important and should be presented within the tables, or better the table, as mentioned above.
We changed VT/VF to life-threatening VA event and HF to HF-related event. Primary and secondary endpoints are now consistent within tables 5 and 6 and the methods section.
17. Limitations: ancestry groups: are other data/literature similar?
We found a single study on a paroxysmal atrial fibrillation in a population in a Japanese population. In that study the authors found an allele frequency of 26%, which is similar to population data of East Asians as found in gnomAD. However, this allele frequency is lower compared to European population (around 40% compared to 26%). On the other hand, population data of South Asian ancestry are found with an allele frequency of 45%. Therefore, we tried to clarify this statement in the limitations, see lines 690-695.
